

# Experience and coping strategies of bowel dysfunction in postoperative patients with rectal cancer: a systematic review of qualitative evidence

Zhang Yanting[1],[*], Dandan Xv[1],[*], Wenjia Long[2], Jingyi Wang[2], Chen Tang[2], Maohui Feng[2], Xuanfei Li[2], Bei Wang[1] and Jun Zhong[2]

[1] Department of Nursing, Wuhan University Zhongnan Hospital, Wuhan, China
[2] Department of Gastrointestinal Surgery, Zhongnan Hospital of Wuhan University, Wuhan, China
[*] These authors contributed equally to this work.

Corresponding authors
Bei Wang, 1745679052@qq.com
Jun Zhong, 1881919@qq.com

## ABSTRACT

**Aim**. Due to the changes of bowel physiological structure and functional disorders after rectal cancer surgery, patients will face many bowel dysfunction for a long time, which will greatly affect their quality of life. The purpose of this review is to integrate the qualitative research on the experience of bowel dysfunction and coping strategies in postoperative patients with rectal cancer.

**Methods**. Systematic retrieval of PubMed, EMbase, Cochrane Library, CINAHL, Web of Sciences, PsycINFO, Willey and other databases was carried out by using the method of subject words and keywords. The Critical Appraisal Skill Programme (CASP) Qualitative Studies Checklist was used for Qualitative assessment. The findings were extracted from the included study and synthesized into the final themes, which was evaluated strictly in accordance with the ConQual process.

**Results**. Nine studies involving 345 participants were included and two main themes were extracted: "Experience a series of changes caused by bowel dysfunction" and "nmet needs and coping strategies facing bowel dysfunction". The changes of rectal cancer patients who experience bowel dysfunction after operation mainly include three parts: bowel dysfunction is more than just a bowel reaction, which covers the bowel symptoms themselves and the subsequent body-related symptoms. The interruption of a normal life, mainly reflected in personal, family, and social life. Complex psychological reactions to bowel dysfunction, psychological changes have a dual nature, showing a positive and negative intertwined. There are two main aspects of unmet needs and coping strategies: the demand is mainly manifested in the need for information and support from medical professionals, while the coping strategy mainly includes diet, activity and drug management.

**Conclusion**. Rectal cancer patient often experience persistent bowel dysfunction after operation, which has a certain physical and mental effects. A series of new needs of postoperative patients are often not fully met, and patients often rely on their own empirical attempts to seek balance, less can get professional support. Future studies need to focus on how to provide continuous information support for postoperative rectal cancer patients, especially professional care from health care staff.

## INTRODUCTION

Rectal cancer is a common malignant tumor in the world, which brings a huge cancer burden to global health (*Bray et al., 2018*). By 2020, rectal cancer was the third most common cancer, accounting for 10% of the estimated new cancers worldwide, and cancer-related deaths ranked second (9.4%) (*Sung et al., 2021*). Surgery is the cornerstone of curative treatment, so surgery has become the main way of radical treatment of rectal cancer, And the main treatment method of early colorectal cancer is surgical resection, including fiber electronic endoscopic resection, local resection and local colorectal resection (*Dekker et al., 2019*). The location and size of rectal tumor growth affect whether the anal sphincter can be preserved. In the past, many patients often need to remove the sphincter and perform permanent colostomy. With the rapid development of low colorectal diagnosis and treatment and bowel anastomosis, the current focus is on the technique of preserving sphincter and avoiding permanent colostomy (*Rouanet et al., 2021*). Anterior resection with sphincter preservation has become the gold standard for the treatment of rectal cancer (*Inoue & Kusunoki, 2010*).

The therapeutic effect of rectal cancer has been significantly improved, and the five-year survival rate of patients has reached 64% (*Miller et al., 2019*). However, patients often experience persistent bowel symptoms and dysfunction after surgery (*Rutherford et al., 2020*). Studies have shown that 90% of postoperative rectal cancer patients and nearly 20–50% of rectal cancer postoperative survivors report varying degrees of bowel dysfunction, such as changes in defecation characteristics, urgent defecation, increased defecation frequency, difficulty in emptying, fecal or urinal incontinence, repeated defecation pain, *etc* (*Ziv et al., 2013*; *Juul et al., 2014*; *Annicchiarico et al., 2021*). Bowel dysfunction after rectal resection and reconstruction is traditionally known as low anterior resection syndrome. These symptoms may improve over time, reaching a stable state after about one to two years, or maybe longer (*Reinwalds, Blixter & Carlsson, 2018*).

With the development of modern medical model, the outcome evaluation of cancer patients is not only the cure rate and survival rate, but also the physical and mental experience (*Firkins et al., 2020*). Postoperative bowel dysfunction has the characteristics of unpredictability and long cycle of treatment and recovery, which greatly affects the quality of life of patients (*Cabilan & Hines, 2017*; *Qaderi et al., 2021*). In view of the privacy of bowel symptoms, many patients may not be willing to ask initiatives for help. A survey of 101 rectal cancer patients who underwent sphincter sparing surgery found that 71.3% reported changes in defecation habits after surgery, but less than 50% of patients actively reported symptoms (*Nikoletti et al., 2008*). Rectal cancer experts are different from patients in understanding the symptoms of bowel dysfunction, as they underestimate the impact of the aggregation and urgency of bowel symptoms (*Chen, Emmertsen & Laurberg, 2014*; *Desnoo, 2006*).

The changes of bowel function have a great impact on the daily life and psychosocial status of patients (*Lu, Huang & Chen, 2017*). Early postoperative patients experience pain and vulnerability due to significant changes in bowel function, and often adopt conservative strategies due to poor symptom management (*Taylor & Bradshaw, 2013*). The treatment of bowel symptoms often depends mainly on patients' self-management, which were often based on their own repeated attempts and lack understanding of the occurrence and evolution of symptoms (*Landers, Mccarthy & Savage, 2012*). *Pape et al. (2021)* found that active follow-up nursing strategies are important for the management of bowel symptoms. In recent years, some researchers have begun to pay attention to the diet management (*Liu et al., 2021b*). Healthy diet programs tend to lead to better physical and role function and less fatigue (*Kenkhuis et al., 2021*). Patients with or without an ostomy were found to undergo a large number of persistent adjustments, and it was difficult to find a modulated management strategy due to the unpredictability of bowel function (*Sun et al., 2015*). *Liu et al. (2021a)* pointed out the necessity of systematic, scientific and continuous guidance of dietary behavior, as well as the management strategies and emotional support. *Van der Heijden et al. (2018)* proposed measures to strengthen early screening of bowel symptoms and supportive care after discharge.

Although the postoperative bowel symptoms of rectal cancer can be evaluated by objective indicators (*Chen, Emmertsen & Laurberg, 2015*), individual differences may affect the understanding of intestinal symptoms. Qualitative research is based on in-depth mining of small sample groups to obtain information, the validity and extensibility of single research results are still limited. All aspects of postoperative bowel symptom experience need to be integrated based on multiple qualitative studies to form stronger evidence. Our purpose is to describe the perioperative experience and needs of patients with rectal cancer experiencing bowel dysfunction, and summarize their feelings and responses by integrating relevant qualitative studies. The results of this study can provide a reference for nurses to formulate practical measures to implement bowel function management. For example, it can make nurses pay more attention to the psychological experience and lifestyle of patients after intestinal surgery, and help to tutor family members, especially their spouses, about their emotional changes and sexual life, so that patients can better return to social life and socialization.

## METHODS

The purpose of this study was to integrate qualitative research on the experience of bowel dysfunction and coping styles of postoperative patients with rectal cancer. Meta integration is a method to collect, understand, compare, analyze and summarize the results of qualitative research on a particular phenomenon, so as to integrate into a new comprehensive explanation, in order to have a more in-depth understanding of the phenomenon (*Sidani, 2008*). System and the review protocol was registered in PROSPERO International prospective register of systematic reviews (ID = CRD42021277878).

In addition, the review was produced in accordance with the Enhancing Transparency in Reporting the Synthesis of Qualitative Research (ENTREQ) statement (*Tong et al., 2012*).

## Inclusion criteria

Studies that reported the experience and coping with bowel dysfunction after rectal cancer surgery using any of the following qualitative data collection methods (interviews, focus groups, or other responses) were eligible for inclusion. There are no restrictions on research design, qualitative research using phenomenology, grounded theory, descriptive analysis, ethnographic research, action research and other theories as research methods will be included. Studies that are not available in full text or whose data are incomplete, duplicated, or not published in English will be excluded.

## Search strategy

The questions in this study include: "what is the experience of intestinal symptoms in postoperative patients with rectal cancer?" "what are the coping strategies for postoperative rectal cancer patients with intestinal symptoms?" . Based on the above research problems, systematic search is carried out on PubMed, Embase, Cochrane Library, CINAHL, web of Sciences, PsycINFO and Wiley databases. The retrieval time is from the time of building the database to October 2021. The following subject words and keywords were used: "rectal cancer", "intestinal symptoms", "patient experience" and "qualitative study". The references in the study were also evaluated to ensure that all relevant studies were included, and the search procedure is shown in Fig. 1.

## Study selection

Import the research retrieved from each database into NoteExpress and delete the duplicate studies. Two researchers (WL and DX) who had received evidence-based training and learning conducted literature retrieval independently, excluding articles that irrelevant with the subject. Two researchers independently read and analyzed the topics and abstracts, traced the references, and again excluded the literature that did not meet the inclusion criteria. If there are differences in the process of study extraction, the third researcher (JZ) will assist in the review to reach a consensus. After the initial screening, the methodological quality of the remaining studies was evaluated, the full text of the included study was obtained, and the included study was finally determined. Figure 1 illustrates the search and screening process

## Assessment of methodological quality

The literature quality was evaluated independently by two researchers (WL and DX), and the third researcher (JZ) decided if there were any differences. In order to ensure the quality of the research results, all selected papers were methodologically evaluated using the Critical Appraisal Skill Programme (CASP) Qualitative Studies Checklist (CASP, 2017) to ensure that they reported all the relevant details of their methodology and analytical methods. All items in the list are listed as "Yes", "No" or "unclear". All items on the list were excluded as poorly rated studies because they did not meet the predetermined eligibility criteria. Studies rated B level or above were included and are reflected in the results and conclusions of this review by extracting and synthesizing the findings. A total of 9 studies were included in this systematic review, all of which were rated as B level. The main weakness of these studies lies in the absent a clear description of the positioning of

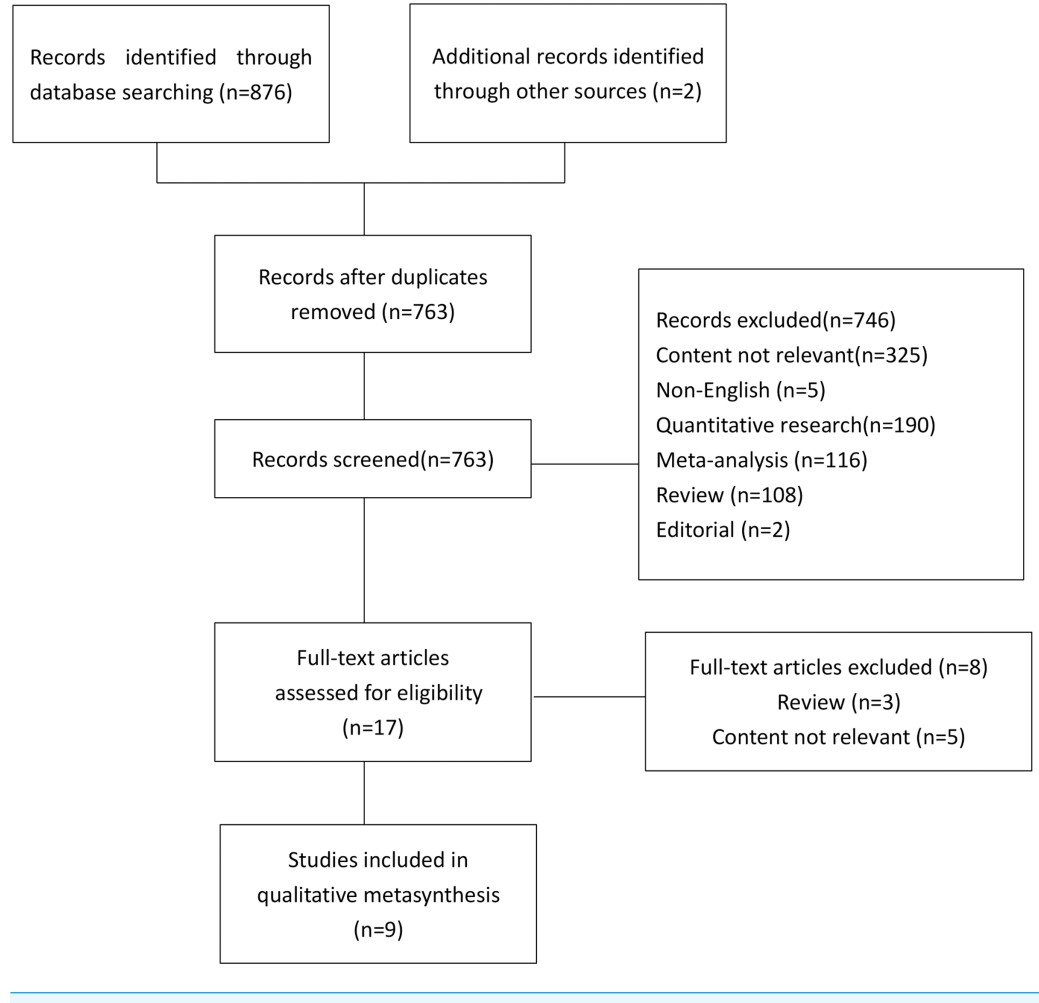

**Figure 1  Literature review strategy.**

researchers and their impact on participants. In-depth discussion of participants' ideas and views in culture or values is insufficient. In addition, the discussion based on theory and philosophy is also lacking. The results of literature quality evaluation are shown in Table 1.

## Data abstraction and analysis

Qualitative data are extracted by two independent examiners (WL and DX) using a unified standardized data extraction format and imported into EXCEL for collation. The extracted data include year, country, research design, participants, phenomena of interest and main results in Table 2. The analysis and collation of qualitative data includes three steps. In the first step, after careful reading and study, the main categories and topics are extracted by two researchers. Each finding was independently evaluated by two researchers, and all findings were categorized into one of the following classifications: unequivocal (evidence beyond reasonable doubt); credible (contains illustrations that may be challenged); or unsupported (when findings are not supported) (Table 3) (*Munn et al., 2014*). The second

**Table 1  Methodological quality of included studies (*n* = 9).**

| Reference | Q1 | Q2 | Q3 | Q4 | Q5 | Q6 | Q7 | Q8 | Q9 | Q10 | Level |
|---|---|---|---|---|---|---|---|---|---|---|---|
| *Desnoo (2006)* | Y | Y | Y | Y | Y | U | U | Y | Y | Y | B |
| *Landers, Mccarthy & Savage (2012)* | Y | Y | Y | Y | Y | U | U | Y | Y | Y | B |
| *Taylor & Bradshaw (2013)* | Y | Y | Y | Y | Y | U | U | Y | Y | Y | B |
| *Sun et al. (2015)* | Y | Y | Y | Y | Y | Y | U | Y | Y | Y | B |
| *Lu, Huang & Chen (2017)* | Y | Y | Y | Y | Y | Y | U | Y | Y | Y | B |
| *Reinwalds, Blixter & Carlsson (2018)* | Y | Y | Y | Y | Y | U | U | Y | Y | Y | B |
| *CASP (2017)* | Y | Y | Y | Y | Y | U | U | Y | Y | Y | B |
| *Van der Heijden et al. (2018)* | Y | Y | Y | Y | Y | U | U | Y | Y | Y | B |
| *Liu et al. (2021a)* | Y | Y | Y | Y | Y | U | U | Y | Y | Y | B |

Notes.

N, no; U, unclear; Y, yes; Y, yes, indicates a clear statement appears in the paper which directly answers the question.; N, no, indicates the question has been directly answered in the negative in the paper.; U, unclear indicates there is on clear statement in the paper that answers the question or there is ambiguous information presented in the paper.

Critical appraisal questions for qualitative studies:

1, Is there congruity between the stated philosophical perspective and the research question or methodology? 2, Is there congruity between the research methodology and the research question or objectives? 3, Is there congruity between the research methodology and the methods used to collect data? 4, Is there congruity between the research methodology and the representation and analysis of data? 5, Is there congruity between the research methodology and the interpretation of results? 6, Is there a statement locating the researcher culturally or theoretically? 7, Is the influence of the researcher on the research, and vice-versa, addressed? 8, Are participants, and their voices, adequately represented? 9, Is the research ethical according to current criteria or, is there evidence of ethical approval by an appropriate body? 10, Do the conclusions drawn in the research report flow from the analysis, or interpretation of the data?

step is two researchers integrating the results and, if necessary, a third researcher stepping in to reach consensus. The third step is to make a new interpretation of the establishment of new categories and themes.

## Assessing certainty in the findings

The final extracted findings were graded using the ConQual method, and the reliability or credibility of each included study was evaluated according to the value and evidence level of the study (*Munn et al., 2014*), as shown in Table 4. All comprehensive results start with a high score, followed by an assessment of reliability and credibility. If there are problems in terms of reliability or credibility, the findings will be downgraded.

# RESULTS

The research included in this study (*n* = 9) includes four main qualitative research methods: phenomenology, grounded theory, prospective research and descriptive research. These studies were conducted in England (*n* = 2), Sweden (*n* = 2), China (*n* = 2), as well as Ireland, the United States and the Netherlands. This review included 345 postoperative rectal cancer patients and identified two comprehensive findings: experience a series of changes caused by bowel dysfunction and the unmet needs and coping strategies facing bowel dysfunction.

## Synthesized finding 1: experience a series of changes caused by bowel dysfunction

The first theme is related to the physical and psychological experience of postoperative patients with rectal cancer. The main concern is the feeling and experience of postoperative

Yanting et al. (2023), *PeerJ*, DOI 10.7717/peerj.15037

**Table 2  Characteristics of included studies.**

| Author (Year) | Country | Design | Theory | Participants | Setting | Phenomena of interest | Main categories |
|---|---|---|---|---|---|---|---|
| *Desnoo (2006)* | England | Qualitative research | Grounded theory | 27 patients who underwent rectal cancer surgery at an interval of one year and completed stoma reversal surgery for at least 6 months | A NHS Trust hospital. | To explore how individuals recovered and adapted following surgical resection of their rectal cancer and the syndrome that occurs as a consequence of the operation. | Three categories were identified: (1) Adapting to the physical changes; (2) Psychological adaptation; (3) Stigma. A secondary theme, the feeling of confidence and normality running through out all these categories. |
| *Landers, Mccarthy & Savage (2012)* | Ireland | A semi-structured question that formed part of a larger multi-site quantitative correlational study. | The symptom management theory | 123 patients who underwent sphincter saving surgery for rectal cancer in the past 3–42 months. | 10 sites specialising in colorectal cancer in Ireland. | To explore participants' qualitative perspectives on bowel symptom experiences and management strategies following sphincter saving surgery for rectal cancer. | Symptom experience: (1) Bowel symptom perception; (2) Bowel symptom evaluation; (3) Bowel symptom responses. Self-Care strategies: (1) Functional self-care strategies; (2) Activity related self-care strategies; (3) Alternative self-care strategies. |
| *Taylor & Bradshaw (2013)* | England | Descriptive Phenomenology | Husserl's Phenomenology | 8 patients experience changes in bowel function, following rectal cancer treatment and stoma reversal 4 to 6 weeks. | A single center, a specialist hospital for colorectal cancer. | To explore the lived experience of patients who experience changes in bowel function, following rectal cancer treatment and stoma reversal. | Six themes were identified in this study: (1) Impact on daily life; (2) Unpredictability; (3) Coping emotionally; (4) Coping practically; (5) Perceived support; (6) Expectations. |

**Table 2** (*continued*)

| Author (Year) | Country | Design | Theory | Participants | Setting | Phenomena of interest | Main categories |
|---|---|---|---|---|---|---|---|
| *Sun et al. (2015)* | America | Focus groups and qualitative interviews | Direct qualitative content analysis | 63 survivors participated in focus groups. 30 female CRC survivors accepted individual interviews. | Data pooled from several studies that assessed HRQOL in CRC survivors. | To explore specific strategies used by survivors to manage bowel dysfunction. | Strategies for regulating bowel function: (1) Dietary adjustments: (a) food categories to avoid; (b) specific foods to avoid; (c) helpful foods. (2) Behavioral adjustments: (a) related to meals and eating; (b) non-meal related. (3) Exercise. (4) Medications: (a) anti-diarrheal agents; (b) bulking agents; (c) pain medications. |
| *Lu, Huang & Chen (2017)* | China | Descriptive Phenomenology | Husserl's Phenomenology | 16 post-operative rectal cancer patients with altered bowel function. | A cancer center in Taiwan. | To explore specific strategies used by survivors to manage bowel dysfunction. | Three themes emerged: (1)"Living in the restroom": (a) uncontrolled excrement; (b) unwilling urination. (2) "Never backward": (a) mood struggles; (b) interrupted daily activities; (c) disturbed family life. (3) "Rebalancing on a new road": (a) spiritual reconstruction; (b) new excrement model; (c) adjusted lifestyle; (d) peer support. |

Yanting et al. (2023), *PeerJ*, DOI 10.7717/peerj.15037

**Table 2** (*continued*)

| Author (Year) | Country | Design | Theory | Participants | Setting | Phenomena of interest | Main categories |
|---|---|---|---|---|---|---|---|
| *Reinwalds, Blixter & Carlsson (2018)* | Sweden | Phenomenological study | Hermeneutical phenomenology | 10 participants, 12–20 months after surgical closure of a temporary loop-ileostomy following rectal cancer surgery. | One public university hospital and one public county hospital in Sweden. | To illuminate what it means to live with a resected rectum due to rectal cancer, after reversal of a temporary loop-ileostomy. | Three themes were identified in this study: (1) Living with uncertainty; (2) Struggling to live with altered bowel function; (3) Preoccupation. Bowel function. |
| *(CASP)* | Sweden | Explorative qualitative design based on narrative interviews | Qualitative content analysis | 16 participants included 9 women and 7 men who had undergone surgery for rectal cancer with an anterior resection and received a temporary loop ileostomy | One public university hospital and one public county hospital in Sweden. | To describe the first 4 to 6 weeks after reversal of a temporary loop ileostomy due to rectal cancer. | Three themes were identified in this study: (1) Life being controlled by the altered bowel function; (2) Striving to regain control over the bowel; (3) A desire to be normal. |
| *Van der Heijden et al. (2018)* | Netherlands | Focus group sessions | Inductive content analysis | 16 patients (males = 50%) who had treated with a low anterior resection for rectal or distal sigmoid malignancy. | A non-academic Dutch teaching hospital | This study aimed to explore the impact of LARS from a patient perspective facilitating the construction of a set of recommendations improving current care stratagems. | Three themes were identified: (1) illness perception; (2) preoperative care; (3) postoperative supportive care. |
| *Liu et al. (2021a)* | China | Descriptive qualitative | System management theory | 36 participants were between 2 months and 2 years after having sphincter-saving surgery. | A public university hospital | To explore the experience of dealing with defecation dysfunction by changing the eating behavior of people with rectal cancer following sphincter-saving surgery | Three themes were identified in this study: (1) Have motivations to change diet; (2) Need strategies to change diet; (3) face barriers to change diet. |

**Table 3  Findings extracted from the included studies with illustration.**

Desnoo, L., & Faithfull, S. (2006). A qualitative study of anterior resection syndrome: the experiences of cancer survivors who have undergone resection surgery European journal of cancer care, 15(3), 244–251. https://doi.org/10.1111/j.1365-2354.2005.00647.x

| | |
|---|---|
| Finding | The interruption of the daily rhythm (C) |
| Illustration | "…in the beginning we did find it difficult, we didn't like going anywhere…it did take a couple of months before we did go out. I'm still hardly 'normal', I mean what I would call normal is going to the loo after breakfast, you know, how we were brought up as children." |
| Finding | Positive psychological adaptation (C) |
| Illustration | "…it certainly hasn't worried me because I just feel lucky to be alive, so what ever I have to cope with, it doesn't really matter. It's part of life. You just have to accept it and not dwell on it." "...you do get frightened, you think is this normal. |
| Finding | Share with your peers (C) |
| Illustration | "when you have common ground you're prepared to exchange these very personal things." |
| Finding | Form a new interpretation (C) |
| Illustration | "It's solid now [bowel consistency] except when something upsets me…I know that bananas upset me." |

Landers, M., McCarthy, G., & Savage, E. (2012). Bowel symptom experiences and management following sphincter saving surgery for rectal cancer: A qualitative perspective. European journal of oncology nursing: the official journal of European Oncology Nursing Society, 16(3), 293–300. https://doi.org/10.1016/j.ejon 2011.07.002

| | |
|---|---|
| Finding | Differences in severity of bowel symptoms (C) |
| Illustration | "The most difficult is dealing with incontinence," (Participant: 37). "Constipation is the most problematic symptom" (Participant: 47). "I have discomfort in the back passage at almost all times and this can become very sore during bowel movement especially if somewhat liquidity"(Participant: 89). |
| Finding | Symptoms are unpredictable (C) |
| Illustration | "when I get diarrhoea it lasts 2 days. I usually get it once a fortnight" (Participant: 19)." |
| Finding | An attempt at self-care (U) |
| Illustration | "the main issues are that I can go a few times daily and it is very hard to regulate it. My movements are very small & I could have 6–7 of these in 1 h. Also I get sore from using toilet paper and find it better to shower after movements". (Participant: 24). |
| Finding | Lose confidence (C) |
| Illustration | "the unpredictability (sic) has been most problematic for me, sometimes certain foodstuffs cause no reaction and other times the same food stuff runs through me. I have no confidence in any food stuff being "risk free"(Participant: 05). |
| Finding | Lucky point of view (U) |

**Table 3** (*continued*)

| | |
|---|---|
| Illustration | "I was one of the lucky people that got over my surgery quickly with little or no side effects of bowel symptoms" (Participants: 59). |
| Finding | Concerns about the prognosis (U) |
| Illustration | "I often wonder if I will have these symptoms for the remainder of my life" (Participants: 21). |
| Finding | Disruption of sleep rhythm (C) |
| Illustration | "occasionally sleep was disturbed because of the need to visit the toilet"(Participant: 99). "…having to go to the toilet a number of times during the night. This affects my sleep patterns and ultimately causes tiredness and fatigue" (Participant: 104). |
| Finding | Diet adjusts (U) |
| Illustration | "in an effort to control flatus avoided fizzy drinks and pulses" (Participant: 15). "When I eat a lot of sweet food –e.g., chocolate, cakes etc –I get diarrhoea so as a result I avoid these foods" (Participant: 13). "I eat a lot of apples, bananas, puddings like rhubarb I find very good. My visits to the toilet can be 3–4 times a day as bowel don't seem to empty but I manage and hope for the best" (Participant: 17). |
| Finding | Use antidiarrheal medicine (U) |
| Illustration | "constant movement of bowel (the) only medication I take is tablets imodium (loperamide). I can manage my symptoms pretty well" (Participant 14). |
| Finding | In the treatment of constipation (U) |
| Illustration | "when constipation occurs I take novicol (Movicol) pres (prescribed) by doctor. I repeat 3–4 days, it is very easy to take and works slowly" (Participant: 18). |
| Finding | Use protective pads (C) |
| Illustration | "I share a house so I have a private store of baby wipes, face cloths and soothing creams so I can always cope with incontinence I sleep with a sanitary pad to avoid soiling" (Participant: 15). "The use of pads when not at home gives peace of mind but I don't always wear one as I have learnt which days are going to be problematic –I can tell once I am up & made my first visit to the toilet" (Participant: 33). |
| Finding | Change activity pattern (U) |
| Illustration | "avoids breakfast to allow for 'safe travel to work" (uses public transport)" (Participant: 11). "…eliminate deadlines and manage schedules" (Participant: 106). |
| Finding | Other treatment options (U) |
| Illustration | "bio-energy therapy, spirituality, self-belief and yoga" (Participant: 21). |

Taylor, C., & Bradshaw, E. (2013). Tied to the toilet: lived experiences of altered bowel function (anterior resection syndrome) after temporary stoma reversal. Journal of wound, ostomy, and continence nursing: official publication of The Wound, Ostomy and Continence Nurses Society, 40(4), 415–421. https://doi.org/10.1097/WON.0b013e318296b5a4

**Table 3** (*continued*)

| | |
|---|---|
| Finding | Fear of going to the bathroom (C) |
| Illustration | "It has restricted my life…because there is always this fear of having to go to the toilet." |
| Finding | A new understanding of bowel function (U) |
| Illustration | "Before this happened, before my diagnosis, I never thought about going to the toilet. Not having control. I now have to wear large pads to control it."<br>"I think the fact I had studied the leaflet in detail before the reversal helped, but my experience was bad, if not worse, than what it says here."<br>"No mention of it was ever made of me, when I came home I thought that the bowel would get back to working…but then I had terrible diarrhea." |
| Finding | Bowel symptoms are difficult to control (C) |
| Illustration | "So long as I can get back to some routine once or twice a day that doesn't bother me then I can get on with my life. So long as I can function normally but when you have to go to the toilet every 5 min …well that bothers me." |
| Finding | Understanding the timing of symptoms (C) |
| Illustration | "I've been a lot worse since having the reversal. And I've questioned it …maybe long-term it's worth having it done. Short-term it's…hell." |
| Finding | Skin pain (C) |
| Illustration | "It [the skin] was very, very sore to such an extent that I was afraid to go to the toilet because it hurt so much." |
| Finding | Flushing helps (U) |
| Illustration | "I take the Loperamide and Codeine Phosphate to control the fragmentation …it's so difficult to get empty…but by douching this helps get me clean or stop getting sore." |
| Finding | The disappointment of getting back to normal (C) |
| Illustration | "I thought it wouldn't last long and settle a bit quicker. I was told there would be slight changes but only for a while. But not like this." |

Sun, V., Grant, M., Wendel, C. S., McMullen, C. K., Bulkley, J. E., Altschuler, A., Ramirez, M., Baldwin, C. M., Herrinton, L. J., Hornbrook, M. C., & Krouse, R. S. (2015). Dietary and Behavioral Adjustments to Manage Bowel Dysfunction After Surgery in Long-Term Colorectal Cancer Survivors. Annals of surgical oncology, 22(13), 4317–4324. https://doi.org/10.1245/s10434-015-4731-9

| | |
|---|---|
| Finding | Lost job (C) |
| Illustration | "An overdose of coleslaw put me out of commission for three months. I lost a job over this, so it is important what you eat." |
| Finding | Adjust the time (U) |
| Illustration | "I just try to balance when I eat things versus what my schedule is gonna be.…you just plan out your schedule and figure out when you're gonna eat what when. I still eat everything that I want. I just don't necessarily get to eat it when I want to eat it."<br>"If I have to change my pouch or I have other things to do, I get as much things done in the morning that I need to do and so even stretch it out a little longer, so by the time I'm ready to change the pouch, I'm ten or twelve hours have gone by between eating." |

**Table 3** (*continued*)

| | |
|---|---|
| Finding | Social avoidance (C) |
| Illustration | I don't…choose to be around groups of people I do not know. I suppose there's times when I've cut myself off from something that might be an enjoyable activity. |
| Finding | Changes in activity intensity (U) |
| Illustration | ''You have to really watch your diet and exercise. You know, walking, exercycle, swimming.'' ''I try not to drive as much as I used to. When I drive, I want to make sure that I know where I'm going.'' |
| Finding | Different drug effects (C) |
| Illustration | ''I've always had to take Imodium and it still doesn't really do a whole bunch of good.'' ''He gave me Metamucil to try and that just kind of makes a sludge.'' ''And even though I have the Imodium, that doesn't help immediately.'' |

Lu, L. C., Huang, X. Y., & Chen, C. C. (2017). The lived experiences of patients with post-operative rectal cancer who suffer from altered bowel function: A phenomenological study. European journal of oncology nursing: the official journal of European Oncology Nursing Society, 31, 69–76. https://doi.org/10.1016/j.ejon.2017.10.004

| | |
|---|---|
| Finding | Mood struggles (C) |
| Illustration | ''I cannot deal with the future …I clean my underwear all day''. (P4) ''It's so excruciation! It's spiritual torture! I cannot go outside. I am jailed at home and frequent to restroom.? (P15) ''The stool passed before I took off my underwear. My dress became dirty! …When I put on my underwear, a little urine dripped out and soiled my dress, too. Although I was alone at home, I felt quite ashamed.'' (P1) ''I pass small pieces of stool for minutes throughout the day. When I put on my underwear, I often feel fecal urgency and have to pass stool immediately. I feel like I'm being played and I'm tied down to the restroom.'' (P15) |
| Finding | Interference with family harmony (C) |
| Illustration | ''Because the amount of time I stay at home has increased, my family and I have more and more arguments about life habits. ''(P10) ''When I go to restroom at midnight, the noise of flushing water often alarms my wife. ''(P10) ''When I have frequent loose stools, the foul odor contaminates my entire body, my clothes, and the restroom. My children are disgusted by the fecal odor. Whenever I'm in the bedroom or living room, they notice me rushing to the restroom and they quickly step aside. Although my wife sometimes has a similar response, she often tolerates it sympathetically!'' (P10) ''We only have one restroom, so I have to pass stool even when a family member is taking bath in there. …My daughter always cannot forgive me. …she is sixth grade student of primary school.'' (P10) |

**Table 3** (*continued*)

| | |
|---|---|
| Finding | Changes in sexual relationships (C) |
| Illustration | "Now I don't have a hard and long-lasting erection sufficient for intercourse. Before the surgery, I enjoyed it once a week! Now, I can't! I give it up! My wife understands my bowel symptoms and the erectile dysfunction." (P4)<br>"When I have sexual desire, I hug my wife. When she has needs, she sometimes comes to hug me, too. She entirely understands me!" (P3) |
| Finding | Confidence in improvement (C) |
| Illustration | "The surgeon tells me, 'You will get better and better'. I noticed the reality and believe that 'I will be better'. "(P6) |
| Finding | Seek peer support (C) |
| Illustration | "I joined a social network and found that we have similar experiences. They totally understand me, so I am not alone." (P10)<br>"I ask other patients for mental support and to share information with each other." (P16)<br>"I go outside with other patients. When I just say, 'I have to do something,' they smile with understanding and know what I'm going through. I don't feel ashamed and don't need to cover up the foul odor. Moreover, we don't even need that much verbal communication!" (P10)<br>"I have severe anal pain related to the small, frequent defecation; other patients advised me to use a cutaneous protective cream which is very useful." (P15) |
| Finding | Religious beliefs (C) |
| Illustration | "I am a Christian. I was initially diagnosed with stage IV disease. My sisters and brothers prayed for me, and then the pathology reported no residual tumor. I believe that it was a miracle from God! So, I believe I will continue to get better." (P10) |
| Finding | Time management for eating (C) |
| Illustration | "I record what I eat, what time I have defecation, how frequent I go to the restroom, and how long the interval of my daily routine is. When I know the association between eating and defecation time, I can control it. When I go outside, make travel plans, and have a dinner party, I know how to arrange it all." (P8) |

Reinwalds, M., Blixter, A., & Carlsson, E. (2017). A Descriptive, Qualitative Study to Assess Patient Experiences Following Stoma Reversal After Rectal Cancer Surgery. Ostomy/wound management, 63(12), 29–37.

| | |
|---|---|
| Finding | Urgency of bowel symptoms (C) |
| Illustration | "Well sometimes I get a kind of urgency that I can't control, that is I have to go…and then it just comes, I don't even have the time to reach the toilet!" Participant 2<br>"Sometimes, suddenly it just disappears and it can be away for a fortnight; I don't need diapers, not anything, and the stool is normal…then all of a sudden (claps his hands together with a bang) it starts again…." Participant 1 |

**Table 3** (*continued*)

| | |
|---|---|
| Finding | Fear of recurrence (C) |
| Illustration | "I did get to see XX (name of the surgeon) in April and then I talked to him and everything looked alright. It was, I was afraid that it would become something more, but he told me that everything looked fine and so on. I shouldn′t worry…But cancer is cancer. It worries…but I have to trust the doctors…" Participant 4 "Every time I go to the toilet I′m reminded that I had cancer! I wonder how it is for others? Not that it is, it doesn′t break me every time you know (laughs), but I've thought about it several times. It′s more like the CANCER is present than that the bowel is different. Because the bowel is different since the cancer has been there…." Participant 9 |
| Finding | The skin gets ulcerated (C) |
| Illustration | "Well the worst is the pain when it gets ulcerated. Because then you don′t really know how to relieve it, since it doesn′t help to wash or use ointments or anything…there have been times when I have been up many, many times at night…" |
| Finding | Change of mindset (U) |
| Illustration | "And…. let's say at the beginning of the year. Then I turned into accepting it more. That…I came to terms with it (laughs) that I accepted, this is it. You have to make the most of it." Participant 4 |
| Finding | New Bowel patterns (C) |
| Illustration | "No, but it does not entirely, er, work as before. Previously, you never thought about whether you went to the toilet or not. It was just something you did, but now the bowels are not so fond of the new situation…" Participant 8 |
| Finding | Social awkwardness (C) |
| Illustration | "Ahhh! I stood at the counter in the store and I just felt help, no! It just came! And I shoved my things away and headed out. Then it was diarrhea! It ran down my legs and I was wearing pale pants. And I met a woman and she looked and I thought, yes, well, let her look I cannot help it!…. Such stuff is so embarrassing!" Participant 3 |

Reinwalds, M., Blixter, A., & Carlsson, E. (2018). Living with a resected rectum after rectal cancer surgery-Struggling not to let bowel function control life. Journal of clinical nursing, 27(3-4), e623–e634. https://doi.org/10.1111/jocn.14112

| | |
|---|---|
| Finding | Social life blows (C) |
| Illustration | "My social life has been severely affected! I say no to all social events. " Participant 5 |
| Finding | Refuse to try new foods (U) |
| Illustration | "If something special's going on then it's better not to eat. I daren't eat in that case because it feels safer to just completely refrain. " Participant 15 |
| Finding | Medical support helps (U) |
| Illustration | "Some tips on how to eat, or that I could have taken that loperamide could have helped me from the start when I had diarrhea at the hospital. So perhaps I wouldn't have had those worst 2 weeks there anyway. That would have been [appreciated]! " Participant 3 |
| Finding | Regret about the operation (U) |

**Table 3** (*continued*)

| | |
|---|---|
| Illustration | "I don't really accept this thing with sitting on the loo for 5 h every day! It doesn't work for me, I can't have a life like that!... I take more of an attitude that my body has to adjust to what I think is right…" Participant 4<br>"I thought it would be completely different. That I'd be able to wear those small briefs again…that it would be like before, but it never is…" Participant 10<br>"Yes, first I thought I would be ecstatic to get rid of the stoma, but I never was! But that was because this was so much trouble instead…" Participant 13 |
| Finding | Spouse support (C) |
| Illustration | "My wife, of course…We know each other inside out and then, when you live together with the problems that one or the other has, and we're very close, so we live with this to-gether…this is what I need…she helps and supports me all the way…" Participant 7 |
| Finding | Lack of medical support (U) |
| Illustration | "I think you can get quite lonely after leaving the hospital…When I was released, I went out through a door and then I was completely alone. I was completely abandoned! I should have had much, much more support than I did…No one can manage by themselves! They can't…" Participant 10 |
| Finding | Luck and Gratitude (C) |
| Illustration | "Oh God, I've been through cancer surgery –what should I expect? I've been lucky! I think I've got a little handicap and I'll have to live with that…I've been given a second chance in life. And I'm going to take good care of it! " Participant 11<br>"Yes, I'm still thankful. That's how I think in order to handle the situation…there's so much else when you look and listen and see that is so much worse…like the opposite to life…" Participant 8 |

van der Heijden, J., Thomas, G., Caers, F., van Dijk, W. A., Slooter, G. D., & Maaskant-Braat, A. (2018). What you should know about the low anterior resection syndrome - Clinical recommendations from a patient perspective. European journal of surgical oncology: the journal of the European Society of Surgical Oncology and the British Association of Surgical Oncology, 44(9), 1331–1337. https://doi.org/10.1016/j.ejso.2018.05.010

| | |
|---|---|
| Finding | Gain understanding (C) |
| Illustration | "I immediately said to my friends, acquaintances [...] "Sorry if I suddenly disappear from the dining table, then I'm taking a shit". Or when I just walk away in the middle of a conversa-tion …I announced up front that this can happen, and all un-derstood." |
| Finding | Feel lonely (C) |
| Illustration | "I am alone so it's so much harder to unburden myself. When I get home, I first have to call my family to tell them what has happened. And picking up the phone to call them is a barrier in itself."<br>"You can be alone when you're together." |
| Finding | Spouse pressure (C) |
| Illustration | "My wife has suffered more from all this than I did. Person-ally, I get over it very easily." |
| Finding | Bowel symptoms were beyond expectation (U) |

**Table 3** (*continued*)

| | |
|---|---|
| Illustration | "I was not thinking about what could happen after the surgery. [...] I thought that they would remove the cancer, and it would be over. But that turned out differently ..." |
| Finding | Information acquisition preference (U) |
| Illustration | "I prefer face-to-face contact I think. It is easier to absorb than written information." "A: I'm not modern enough yet. B: Well, maybe I am, but I am very fond of personal conversations. Then you get the chance to ask your questions directly." "Maybe an educational film is the ideal solution? [...] And not everybody has that level of reading skills." |
| Finding | Preoperative information and decision making (U) |
| Illustration | "A doctor must decide. I don't think you are in a position to do it yourself." "Not for me, because I don't know what I can expect yet." "I would have liked that. I went to the pelvic floor therapy afterwards, but if I had done all this before the surgery, it would have been better for me." "At first, you want the surgery to be over. I can imagine doing it after surgery if you have a stoma. Because you will have to prepare for when the stoma is reversed." "I couldn't say this up front (willingness to participate in preparatory programs), because I didn't know at that moment how I would end up." |
| Finding | Attitude to Exercise (U) |
| Illustration | "I absolutely believe in the added value of the whole process, [...] from the moment you get your stoma until reversal. It's a good thing to exercise your sphincter muscles." |
| Finding | Medical support needs (C) |
| Illustration | "I felt really bad soon afterwards (discharge). The period until you return for your first follow-up appointment feels very long." "I was discharged from the hospital, and I asked that lady: "What do I do now, is there a procedure?"" "You need more information because everyone around you is a lay person; you can't rely on their advice." |
| Finding | Lack of advance notice (U) |
| Illustration | "Reassurance that everything is normal would make a big difference. In fact, that is a part of the preparation process I think. They could inform you about these [postoperative symptoms] in advance!" |
| Finding | Evaluation Tool Support (U) |
| Illustration | "I did not think that I had many complaints. Eventually, they (reference to the colorectal care nurse) came up with a list. [...] I said: "maybe one and a half hours a day on the toilet is quite a long time after all". |

Liu, W., Xu, J. M., Zhang, Y. X., Lu, H. J., & Xia, H. O. (2021). The experience of dealing with defecation dysfunction by changing the eating behaviours of people with rectal cancer following sphincter-saving surgery: A qualitative study. Nursing open, 8(3), 1501–1509. https://doi.org/10.1002/nop2.768

| | |
|---|---|
| Finding | Physical burden (U) |

| | |
|---|---|
| Illustration | "The defecating process was so torturous and unbearable, my anus was always experiencing intense pain. So, I was always trying different foods to facilitate the formation of stool. I have tried bananas, many more starches and whole grains, hoping to find a way to make defecation smooth."(C22). |
| Finding | Trial-and-error approach (U) |
| Illustration | "I have tried several times to determine that watermelon may increase the frequency of defecation once I eat it. I had to decrease the amount of watermelon that I eat."(C14). "corn was determined to be good for my defecation; once I ate corn, my stool seemed to be more likely to take shape."(C7) |
| Finding | Seek medical support (U) |
| Illustration | "I asked the doctors how to eat both in the ward right after surgery and at the clinic and I just follow as their guides now."(C11). |
| Finding | Find coping strategies (U) |
| Illustration | "I found help from traditional medicine about how to relieve my constipation after surgery. The doctor prescribed a Chinese medicine for me and gave me some dietary suggestions; later, it seemed that the constipation was relieved a little, but I was not sure whether it was because of the Chinese medicine's effects."(C27). "The defecation was so unbearable that I searched the internet to learn how to eat; vegetables were suggested to be good for resuming normal defecation and they should be eaten more often." (C11). |
| Finding | Conflicting information (U) |
| Illustration | "Eating more foods high in fiber was suggested by my doctor, so I tried Chinese chives and my defecation was more normal after I took them. However, some wardmates have said that Chinese chives are thought to be stimulating foods in traditional Chinese medicine that may have negative effects on tumors. So, can I go on taking it in the future?" (C29) |
| Finding | Affect family relationship (U) |
| Illustration | "I was eating meals at my relative's house with many people together and my relative was preparing many dishes that including too much meat. I couldn't criticize her because she cooked for us out of kindness, but I didn't mean to eat so much greasy food in case it caused too much defecation."(C24). |
| Finding | Excessive dietary management (U) |
| Illustration | "The more foods I ate, the more times I had to go to the toilet, so I tried to eat less; I even tied to drink the least of amount of water possible." (C24) |

**Table 4  ConQual summary of findings.**

Systematic review title: Experience and coping strategies of bowel dysfunction in postoperative patients with rectal cancer:a systematic review of qualitative evidence
Population: postoperative patients with rectal cancer
Phenomena of interest: the exposition of bowel dysfunction and coping strategies in postoperative patients with rectal cancer
Context: the experience and countermeasures of rectal cancer patients in the postoperative stage

| Synthesized finding | Type of research | Dependability | Credibility | ConQual score |
|---|---|---|---|---|
| Experience a series of changes caused by intestinal dysfunction. Bowel dysfunction is not just an intestinal reaction. The interruption of a normal life. Complex psychological reactions to bowel dysfunction. | Qualitative | Downgrade one level[*] | No change | Moderate |
| Unmet needs and coping strategies facing intestinal dysfunction. Emerging unconsidered unmet demand. Self-management strategy | Qualitative | Downgrade one level[*] | Downgrade one level[**] | Low |

Notes.

[*]Downgraded one level due to common dependability issues across the included primary studies (the majority of studies had no statement locating the researcher and no acknowledgement of their influence on the research).

[**]Downgraded one level due to a mix of unequivocal and equivocal findings.

intestinal dysfunction in postoperative patients with rectal cancer. The core concepts that make up the theme include: bowel dysfunction more than just a bowel reaction; the interruption of a normal life; and complex psychological reactions to bowel dysfunction.

### Category 1.1: bowel dysfunction more than just a bowel reaction

Most participants experienced more than one bowel symptom, the most common of which included fecal incontinence, intestinal urgency, flatulence, diarrhea, constipation, and inadequate evacuation (*Reinwalds, Blixter & Carlsson, 2018*; *Desnoo, 2006*; *Lu, Huang & Chen, 2017*; *Taylor & Bradshaw, 2013*; *Landers, Mccarthy & Savage, 2012*; *Liu et al., 2021a*; *Van der Heijden et al., 2018*; *Reinwalds, Blixter & Carlsson, 2017a*). And these symptoms have special meaning for different individuals, maybe acute symptoms to some people, and may not be so important to others (*Landers, Mccarthy & Savage, 2012*). Many participants expressed a sense of helplessness about uncontrollable symptoms, described the sudden onset and disappearance of bowel symptoms, and said it was difficult to find patterns and signs (*Landers, Mccarthy & Savage, 2012*). However, participants held different views on the urgency of different symptoms (*Reinwalds, Blixter & Carlsson, 2017b*). Consistent with the fact that most participants experienced bowel dysfunction after surgery, they felt uncertain about the occurrence and development of the symptoms (*Reinwalds, Blixter & Carlsson, 2018*; *Lu, Huang & Chen, 2017*; *Taylor & Bradshaw, 2013*; *Landers, Mccarthy & Savage, 2012*; *Reinwalds, Blixter & Carlsson, 2017a*). While experiencing bowel symptoms, participants often mention the time characteristics of the symptoms, and the participants often regard the regularity of the occurrence time of the symptoms as the effective control of the symptoms (*Lu, Huang & Chen, 2017*; *Landers, Mccarthy & Savage, 2012*; *Sun et al., 2015*; *Reinwalds, Blixter & Carlsson, 2017a*). Participants who experienced bowel symptoms day and night tended to express concern about the persistence of the symptoms (*Landers,*

*Mccarthy & Savage, 2012*). Some participants reported the bowel dysfunction exceeded their preoperative expectations and cognition, often resulting in deeply personal experiences (*Reinwalds, Blixter & Carlsson, 2018*).

In addition to the discomfort caused by bowel dysfunction, the participants also suffered other physical discomfort in their daily life. Due to a series of chain reactions caused by bowel dysfunction, participants need to go in and out of the toilet frequently and have to interrupt sleep (*Lu, Huang & Chen, 2017*; *Landers, Mccarthy & Savage, 2012*). Long-term work and rest disorders lead to lack of energy and fatigue (*Landers, Mccarthy & Savage, 2012*). In addition, frequent bowel and local skin pain often make participants fear the next time they go to the toilet and lack an effective coping mechanism (*Landers, Mccarthy & Savage, 2012*). Some participants said that they need to maintain a fixed posture due to symptoms caused by bowel dysfunction, which often leads to low back pain, and when they encounter sudden symptoms (diarrhea), they need to change their posture in a short period of time, changing posture often brings unbearable pain (*Reinwalds, Blixter & Carlsson, 2018*).

### Category 1.2: the interruption of a normal life

It is often difficult for participants to return to a normal pace of life after operation, and bowel symptoms have a destructive effect on daily life (*Desnoo, 2006*; *Lu, Huang & Chen, 2017*; *Taylor & Bradshaw, 2013*; *Landers, Mccarthy & Savage, 2012*). Destructive bowel symptoms lead to disruptions in the normal pace of life, and even the daily plans of participants often depend on toilet habits (*Reinwalds, Blixter & Carlsson, 2017a*). Because eating is often closely related to intestinal movement, participants tend to spend more time dealing with intestinal events (defecation) after eating. Participants said that time of eating and the type of food needed to be adjusted after operation to reduce interference with daily life (*Lu, Huang & Chen, 2017*). Some participants said that due to the need for frequent bowel emptying at night, it was difficult to ensure effective rest at night and had to give up work (*Reinwalds, Blixter & Carlsson, 2018*). In addition, there are some participants trying to find a balance between bowel dysfunction and life by developing plans to deal with unpredictable events, such as setting deadlines and schedule management (*Landers, Mccarthy & Savage, 2012*).

Bowel dysfunction also affect the family life of participants, they often feel very difficult to carry out normal family life in the initial stage. Some participants also said that family life has improved to some extent with the passage of time, but it is still need spend more time and energy on planning (*Reinwalds, Blixter & Carlsson, 2017a*). In addition, participants mentioned the modification of household facilities, such as by adding toilets to avoid interference with the family's need to go to the bathroom (*Taylor & Bradshaw, 2013*). The participants also had different attitudes towards the care and support of their close family members. Some participants said that it takes courage to tell their families about the disease, and they often find it difficult to act (*Van der Heijden et al., 2018*). Although participants with partners get more support and care to some extent, they may also face difficulties in opening themselves and having conflicts or quarrels. The feelings and experiences of spouses are often ignored (*Van der Heijden et al., 2018*). Maintain good

communication and understanding, to a certain extent, play a positive role in promoting the relationship between husband and wife, receiving psychological counseling may bring some good suggestions (*Lu, Huang & Chen, 2017*). Sex life is rarely mentioned, and mutual understanding and support between partners is very important (*Lu, Huang & Chen, 2017*).

The unpredictability of bowel dysfunction can easily embarrass participants in their normal social life, and participants often take evasive measures to avoid possible embarrassment (*Landers, Mccarthy & Savage, 2012*; *Reinwalds, Blixter & Carlsson, 2017a*). Participants were afraid of going out to socialize, afraid of sudden intestinal events (*Desnoo, 2006*; *Landers, Mccarthy & Savage, 2012*). Participants often feel difficult and embarrassed to talk about bowel problems, often use euphemisms or special pronouns, and are more willing to share with people with similar experiences (*Desnoo, 2006*). However, some participants said that it was easier to say it than to cover it up, and it was more understandable to others, avoiding the misunderstanding caused by sudden absence due to bowel emergencies (*Van der Heijden et al., 2018*). Peer support among participants has a positive impact on social life. Many participants said that peer support was more comforting, especially suggestions from peer support were often helpful (*Lu, Huang & Chen, 2017*; *Liu et al., 2021a*; *Reinwalds, Blixter & Carlsson, 2017a*).

### Category 1.3: complex psychological reactions to bowel dysfunction

Although the surgery altered the normal structure of their intestines, some participants viewed the changes in gut patterns as a small price to pay for life rather than a bad thing (*Desnoo, 2006*; *Reinwalds, Blixter & Carlsson, 2017a*). Although the participants had a positive attitude towards the recovery of postoperative bowel dysfunction, they realized that their physical condition would never be the same as before the operation, and tried to seek a sense of balance (*Landers, Mccarthy & Savage, 2012*). When participants describe bowel-related problems, they often pursue the meaning and value of life by adopting a relaxed attitude as a strategy. Some participants gave a new explanation to the sudden intestinal time and reconciled themselves with it by naming it a word such as 'ordinary fibrillation' and 'great fibrillation' (*Reinwalds, Blixter & Carlsson, 2017a*). Although there is still persistent bowel dysfunction, participants strive to look for signs of improvement or seek affirmation from doctors to strengthen their confidence in a better outcome (*Lu, Huang & Chen, 2017*). With the passage of time, participants also gradually began to accept bowel dysfunction, the concept changed from negative to acceptance, and even positive response (*Reinwalds, Blixter & Carlsson, 2017a*). There has been little exploration of culture and spirituality, and only some participants mentioned the support provided by religious belief, expecting miracles despite being diagnosed with advanced cancer (*Lu, Huang & Chen, 2017*).

Fear, embarrassment, anxiety, and other bad emotions are often associated with the uncertainty of bowel dysfunction. Participants are often prone to negative emotions or even depression after experiencing frequent and severe intestinal symptoms (*Taylor & Bradshaw, 2013*). Faced with unpredictable and uncontrollable intestinal problems, participants felt lost confidence and even passive avoidance (*Landers, Mccarthy & Savage, 2012*). In addition, some participants felt difficult to judge whether they are normal or not,
which makes them worry about tumor recurrence (*Desnoo, 2006*). After the operation, the participants may develop new bowel patterns, resulting in a new understanding of the 'normal' and 'abnormal' bowel patterns, and may face contradictory psychology. Many participants expressed adaptation and acceptance to colostomy, but after reversal, patients may feel worse under multiple pressures such as expectations and uncertain bowel function (*Taylor & Bradshaw, 2013*; *Reinwalds, Blixter & Carlsson, 2017a*). Some of the participants even expressed regret over the ostomy and even wanted to resume the ostomy after a year or more (*Taylor & Bradshaw, 2013*; *Reinwalds, Blixter & Carlsson, 2017a*).

## Synthesized finding 2: unmet needs and coping strategies facing bowel dysfunction

The second topic is related to the needs and self-management strategies of rectal cancer postoperative participants to deal with bowel dysfunction. The core is that participants gradually realize the impact of bowel dysfunction after operation, try a variety of self-management programs to achieve a balance between life and bowel dysfunction, and then look forward to returning to normal life. The core concepts that make up the theme include emerging unmet needs and self-management strategy.

### Category 2.1: emerging unconsidered unmet demand

Participants often receive a lot of disease-related information before operation, but they pay more attention to the treatment of the disease at this stage. It is difficult to consider the potential problems after operation, and participants do not really get the information they need. For preparation, participants stressed that it was difficult to imagine the impact of postoperative intestinal symptoms on life (*Van der Heijden et al., 2018*; *Reinwalds, Blixter & Carlsson, 2017a*). Some of the participants said that despite being given the information beforehand, they still reported more problems than expected after surgery and more problems after discharge (*Landers, Mccarthy & Savage, 2012*; *Van der Heijden et al., 2018*). In addition, some participants raised the problem of insufficient preoperative hints and the need for adaptive guidance for rehabilitation training before operation (*Van der Heijden et al., 2018*). For some participants who expressed a sense of helplessness, feeling unable to make a judgment or decision, they believed that the decision should be made by the surgeon (*Liu et al., 2021a*; *Van der Heijden et al., 2018*). Most of the participants have a strong demand for relevant information after operation, and the participants also have obvious individual preferences for the way of obtaining information (*Van der Heijden et al., 2018*).

Compared with the period of hospitalization, participants described a feeling of being abandoned and often faced with the problem of access to medical resources after discharge, and it is difficult to obtain effective support from professionals (*Reinwalds, Blixter & Carlsson, 2018*; *Landers, Mccarthy & Savage, 2012*; *Van der Heijden et al., 2018*). After leaving the hospital, participants lack continuous nursing support and are difficult to get professional nursing services, such as colostomy (*Reinwalds, Blixter & Carlsson, 2018*). Some participants indicated that the inadequate discharge procedure and the long interval between the first follow-up (*Van der Heijden et al., 2018*). Furthermore, participants said it was difficult to rely on non-professionals around them and express their expectation

of getting support from medical professionals (*Van der Heijden et al., 2018*). Participants often choose to consult medical staff as much as possible, and some participants say that simple advice from professionals is often very effective (*Liu et al., 2021a*; *Van der Heijden et al., 2018*; *Reinwalds, Blixter & Carlsson, 2017a*). Some participants indicated that their evaluation of their bowel problems was different from that of professionals, and the use of professional assessment tools or checklists was helpful (*Van der Heijden et al., 2018*).

### Category 2.2: self-management strategy

When considering intestinal dysfunction, participants often first attempt to explore the correlation between diet and symptoms, generally dietary adjustments (*Reinwalds, Blixter & Carlsson, 2018*; *Lu, Huang & Chen, 2017*; *Taylor & Bradshaw, 2013*; *Landers, Mccarthy & Savage, 2012*; *Sun et al., 2015*; *Liu et al., 2021a*; *Van der Heijden et al., 2018*; *Reinwalds, Blixter & Carlsson, 2017a*). Participants tried to find food suitable for their bowel function through constant trial and error, and some participants expressed fear of trying new things (*Landers, Mccarthy & Savage, 2012*; *Liu et al., 2021a*). As for the management of eating time, most patients are trying to find a balance (*Lu, Huang & Chen, 2017*; *Sun et al., 2015*; *Liu et al., 2021a*). Participants want to try to coordinate their bowel patterns with the planned schedule and get free eating time as much as possible, but some participants choose more stringent management and eat as little food as possible (*Sun et al., 2015*). Dietary adjustments may have positive results, but when it is difficult to achieve a balance between diet and bowel symptoms, participants may lose confidence in food choices and often choose not to eat foods (*Taylor & Bradshaw, 2013*).

Bowel dysfunctions affect the daily activities of participants, and participants have to change their daily exercise patterns. They may take coping measures such as shortening time, changing exercise patterns, receiving physiotherapy, and wearing protective pads to gain a sense of control and security (*Taylor & Bradshaw, 2013*; *Landers, Mccarthy & Savage, 2012*). Some participants proposed the role of physiotherapy in the control of bowel symptoms and expressed the need for pre-operative contact and understanding (*Munn et al., 2014*). After constantly adapting to regulate the relationship between daily activities and bowel dysfunctions, participants struggled to find a new balance and found personal coping strategies, such as smaller activities, post-exercise showers or the use of diaper pads (*Taylor & Bradshaw, 2013*; *Landers, Mccarthy & Savage, 2012*). In addition, some participants expressed positive views on accepting exercise programs such as bio-energy therapy, spirituality, self-belief, and yoga (*Landers, Mccarthy & Savage, 2012*).

Drugs and related supplements are also often used to help improve bowel dysfunctions, and participants often use drugs tentatively on a doctor's advice or personal experience (*Taylor & Bradshaw, 2013*; *Landers, Mccarthy & Savage, 2012*; *Sun et al., 2015*; *Liu et al., 2021a*). When constipation occurs, although the effect of drugs is slow, it can usually solve the problem better (*Landers, Mccarthy & Savage, 2012*). However, when suffering from diarrhea, taking antidiarrheal drugs can help participants improve the trouble of frequent defecation to a certain extent, but it takes a certain time to take effect, and the effect may vary (*Taylor & Bradshaw, 2013*; *Landers, Mccarthy & Savage, 2012*). Many participants had the problem of poor defecation, which was not effective despite following the doctor's

instructions for the use of drugs (*Sun et al., 2015*), and found that it is more effective to try intestinal irrigation (*Liu et al., 2021a*).

### Assessing certainty in the findings

The comprehensive finding of "Experience a series of changes caused by bowel dysfunction" has been determined to have a reliability problem, so it has been downgraded by one level. in addition, in addition to clear results, there are also some uncertain results, so the ConQual score has been reduced from high to medium. The comprehensive discovery of "Unmet needs and coping strategies facing bowel dysfunction" has the problem of reliability, and it is also found that there is an uncertain part, so its ConQual score is reduced from high to low.

## DISCUSSION

This study describes the bowel and physical symptoms and psychological experiences of postoperative patients with rectal cancer through collective integration. Due to the changes of bowel functional symptoms in postoperative patients with rectal cancer, their original personal life, family life and social life have changed. These changes are a long-term phenomenon, patients need to constantly adjust their lifestyle and attitude to adapt, many patients have a series of negative emotions. Many patients will continue to try to help themselves adapt and accept the changes after rectal cancer surgery by taking drugs or changing their diet. But they do not have systematic and professional support to help them adapt to these changes.

Rectal cancer patients often mention the situation of 'survival mode' before surgery. Patients tend to pay more attention to the treatment-related issues, while the bowel changes and related symptoms that may occur after operation are often not in the primary consideration (*Van der Heijden et al., 2018*). Postoperative rectal cancer patients will have a series of bowel symptoms and physical symptoms, these symptoms will often last for a long time, and even affect the treatment of the patient's disease. The type, frequency and severity of bowel symptoms are affected by diseases, individuals, and other factors. There are certain individual differences, for example, the most disturbing symptom for some patients is diarrhea, while others are constipation or urinary incontinence (*Lu, Huang & Chen, 2017*; *Taylor & Bradshaw, 2013*; *Landers, Mccarthy & Savage, 2012*; *Reinwalds, Blixter & Carlsson, 2017a*). If left untreated, bowel symptoms may cause or increase the pain of patients, and they are unable to return to a normal life (*Landers, Mccarthy & Savage, 2012*). In addition, mental and psychological factors also play an important role during this period. It is necessary to help patients understand the changes of bowel function after operation as soon as possible, and the lack of relevant understanding may lead to high expectations (*Taylor & Morgan, 2011*). Given that symptoms caused by bowel dysfunction are often uncertain, it is difficult for patients to tell whether it is "normal" or "abnormal" and even to worry about tumor recurrence (*Desnoo, 2006*). At present, a scoring tool has been developed to evaluate bowel function (*Emmertsen & Laurberg, 2012*), which can help patients identify their bowel symptoms early. Bowel symptoms may be followed by a range of physical symptoms such as fatigue, sleep disturbances, and pain (*Lu, Huang & Chen,*

*2017*). The patient's physical symptoms are less mentioned and more focused on local skin problems and pain associated with frequent defecation (*Taylor & Bradshaw, 2013*; *Landers, Mccarthy & Savage, 2012*; *Van der Heijden et al., 2018*). Some patients also mentioned that they had to go to the toilet frequently, resulting in sleep interruption and fatigue (*Reinwalds, Blixter & Carlsson, 2018*). Annoying bowel symptoms and worsening physical symptoms can also form a vicious circle, making it necessary to provide more comprehensive support strategies to better promote health outcomes.

The life of patients after rectal cancer has undergone tremendous changes. This process is manifested as the interruption of normal life. The change in the patient's personal life is the first to bear the brunt. The original normal living habits have changed, such as having to go to the toilet frequently, wearing dark clothes, using diaper pads, *etc* (*Reinwalds, Blixter & Carlsson, 2018*). In social life, they will always worry about the risk of incontinence (*Reinwalds, Blixter & Carlsson, 2018*). Frequent use of the toilet and unpleasant smells will make patients deliberately stay away from social activities, which will put them in an embarrassing situation. It's worth noting that despite some stigmatized explanations for bowel-related problems, and perhaps because of this, people with similar experiences are more willing to share them with each other (*Desnoo, 2006*). As the most common caregiver, the partner's importance to the patient's treatment process is self-evident. However, the patient still feels lonely, expressing that it is difficult to open completely, and the caregiver's concerns may also bring pressure (*Van der Heijden et al., 2018*). The sexual lifestyle of patients after rectal cancer surgery has also changed, which is consistent with our study (*Sun et al., 2016*). Bowel symptoms greatly interfere with their daily lives, leading to a range of negative emotions, such as fear, depression, and shame. Kuo's research has similar results (*Kuo et al., 2015*). Negative emotions interact with bowel symptoms, and the uncontrollability of symptoms often leads to more negative behaviors and even unable to get out of the house (*Desnoo, 2006*). Nevertheless, after experiencing negative emotional struggles, most patients expressed confidence in seeking a balance between life and bowel symptoms (*Reinwalds, Blixter & Carlsson, 2018*; *Desnoo, 2006*; *Lu, Huang & Chen, 2017*; *Taylor & Bradshaw, 2013*; *Reinwalds, Blixter & Carlsson, 2017a*). The support of relatives and friends will increase the trust of the person heart, give them comfort (*Van der Heijden et al., 2018*). However, during this period, the patients' negative emotions have not received good attention, and there is still a lack of effective support and intervention mechanisms.

Studies have shown that some patients are unsure of the consequences of surgery, confused about the symptoms they experience, and even regret having the surgery (*Reinwalds, Blixter & Carlsson, 2018*). It's worth thinking about the importance of timing. In the preoperative stage, patients tend to focus on the treatment of the disease, and lack of attention to the possible bowel symptoms after the operation or as an acceptable price (*Desnoo, 2006*). In addition, continuous information and care support after surgery are also very necessary. For patients with rectal cancer, the postoperative recovery period is a long process, and the real challenge is after discharge. Due to the high threshold to contact the hospital, it is difficult for patients to obtain the support of relevant professional medical staff, and they are prone to anxiety (*Van der Heijden et al., 2018*). Different expressions have their own advantages and limitations, and require personal preference. For example, for

some patients, they prefer face-to-face communication rather than written communication. Furthermore, it is very important for patients to get support and help from professionals. Patients are more willing to trust professional medical staff, whose experiential guidance can sometimes provide great psychological support and help (*Liu et al., 2021a*; *Lu & Huang, 2018*).

Burch's research (*2021*) showed that the quality of life of patients after rectal cancer surgery is not high. Pelvic floor muscle function exercise (*Sacomori et al., 2021*), sacral nerve stimulation (*Kuo et al., 2015*), biofeedback training (*Liu et al., 2019*). Which can effectively alleviate patients' bowel symptoms and improve their quality of life, but there are no high-quality studies that show the most appropriate treatment method. This study summarizes how patients after rectal cancer try to use different self-care strategies to deal with bowel symptoms, including functional self-care strategies, activity-related self-care strategies (such as approaching/knowing the location of the toilet), and alternative self-care strategies (such as complementary therapies) and medications (*Landers et al., 2014*). This provides a reference for other patients to manage bowel symptoms. However, dietary changes are often initiated by patients. It is not yet known whether these dietary changes are suitable and universal. Drug management strategies still lack effective consistent plans, and individual differences are large. Therefore, health care providers should evaluate these strategies and help patients evaluate their effectiveness. Patients with rectal cancer usually have a series of symptoms after surgery, so medical staff need to conduct long-term follow-up, assess the actual situation of patients, and provide targeted guidance strategies.

## CONCLUSION

Patients with rectal cancer often experience persistent bowel dysfunctions after surgery, and changes in bowel function have caused tremendous changes. This is accompanied by negative emotional reactions and even a loss of hope for their lives, which seriously affects their quality of life. To find ways to improve bowel dysfunctions, patients will adopt self-care strategies such as diet adjustments, improving activities and use of drugs, but there is still no effective data to prove the rationality and effectiveness of these measures. In addition to the self-regulation of patients, support from family and society is also needed. It is of concern that the professional support provided by health care professionals is consistently highlighted, but further research is needed on how to provide appropriate support services.

### Funding
The authors received no funding for this work.

### Competing Interests
The authors declare there are no competing interests.

## Author Contributions

- Zhang Yanting conceived and designed the experiments, performed the experiments, analyzed the data, prepared figures and/or tables, authored or reviewed drafts of the article, and approved the final draft.
- Dandan Xv conceived and designed the experiments, performed the experiments, prepared figures and/or tables, authored or reviewed drafts of the article, and approved the final draft.
- Wenjia Long conceived and designed the experiments, performed the experiments, prepared figures and/or tables, and approved the final draft.
- Jingyi Wang conceived and designed the experiments, prepared figures and/or tables, and approved the final draft.
- Chen Tang conceived and designed the experiments, performed the experiments, prepared figures and/or tables, and approved the final draft.
- Maohui Feng conceived and designed the experiments, performed the experiments, authored or reviewed drafts of the article, and approved the final draft.
- Xuanfei Li conceived and designed the experiments, analyzed the data, authored or reviewed drafts of the article, and approved the final draft.
- Bei Wang conceived and designed the experiments, authored or reviewed drafts of the article, and approved the final draft.
- Jun Zhong conceived and designed the experiments, performed the experiments, analyzed the data, authored or reviewed drafts of the article, and approved the final draft.

## Data Availability

This is a systematic review of qualitative evidence.

## Supplemental Information

Supplemental information for this article can be found online at http://dx.doi.org/10.7717/peerj.15037#supplemental-information.

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
