# Peer review of "Experience and coping strategies of bowel dysfunction in postoperative patients with rectal cancer: a systematic review of qualitative evidence"

_PeerJ, doi:10.7717/peerj.15037_

## Round 0.1 · original submission · Minor Revisions

Please carefully consider the reviewers' comments before resubmitting.

·

Basic reporting

The grammar and language needs improvements. Though we are able to understand most of the manuscript, there are several places where the meaning is unclear. The authors may benefit from taking the help of proficient English speakers, or professional tools.

Page 7: The authors have mentioned “Surgery has become the main way of radical treatment of rectal cancer”. It will be helpful if the authors could give a short description of how surgery has ‘become’ the main treatment option, and what was the preferred treatment option earlier.

Experimental design

Page 9: The database is names Wiley, not Willey.

The authors can increase the credibility of their study by attaching a filled ENTREQ checklist.

Page 9: It is unclear what the author means by this statement: “All items on the list were excluded as poorly rated studies because they did not meet the predetermined eligibility criteria.”

Page 10: The assessment of certainty in findings that has been explained in section 2.6 should have been split. The authors can keep the methodology in section 2.6, and results of this assessment in the results section.

Validity of the findings

Figure 1: If 746 records have been excluded out of the 760 screened records, shouldn’t 14 records have been assessed for eligibility?

Page 7: “With the development of modern medical model, the outcome evaluation of cancer patients is not only the cure rate and survival rate, but also the physical and mental experience [12].” Corresponding to this, we should remember that patient reported outcomes are now gaining increased attention. The authors may contemplate talking on the use of patient reported outcomes in future studies in the discussion section.

Page 16: Given that you have conducted such an in-depth research into the issue, it could be helpful for the scientific community at large if you can provide some objective pointers into what areas need further research with regard to post-op bowel dysfunction.

Reviewer 2 ·

Basic reporting

The authors provided a systematic review to integrate the qualitative research on the experience of bowel dysfunction and coping strategies in postoperative patients with rectal cancer. The review was mainly about the daily experience and coping strategies. By summarizing related literature, this review pointed out, apart from physical attention, patients need assistance and attention on their lifestyle, mental health, and social life as well. Some of those needs are not yet fully met these days, patients often rely on their own empirical attempts and often lack professional support.
The language is clear, unambiguous, and professional. The authors supplied sufficient background information as well as related references. The review was properly formatted and structured.
However, how the authors presented table 1 can be improved a little. In line 169, the authors mentioned “(Table 1) in Appendix III". However, in the main text, table 1 was not displayed properly at the end of the manuscript. A big blank space was shown. If Table 1 is identical to Appendix III, then it is redundant to put the same material in two places. If not, then the authors need to reformat table 1.

Experimental design

The review was analyzed properly. However, it would be better if the authors can give more professional comments on each coping strategy. Therefore, the review can be more informative to readers. The review goal was well-defined and meaningful. The conclusion was drawn properly based on the analysis. Also, the methods were described in detail.

Validity of the findings

The review pointed out that the psychological health and social life of postoperative patients are crucial and need professional support. The conclusions were well-stated, linked to the original research question, and limited to supporting results.

·

Basic reporting

The article is well-written and the authors have done a good job. Kudos to them.
However, in the introduction section, it is mentioned that one of the objectives of this meta-synthesis is "provide a reference for nurses to formulate practical measures to implement bowel function management." - I feel that is lacking in the recommendation section. A few recommendations can be added to this context e.g. how nurses can help post-opp pts. in immediate coping after surgery, and counseling the family members, especially their spouse regarding their emotional changes, sexual life. etc.

Experimental design

no comment

Validity of the findings

no comment

---

## Round 0.2 · accepted · Accept

Thank you for considering our Journal for your valuable contribution.

·

Basic reporting

no comments

Experimental design

no comments

Validity of the findings

no comments

Additional comments

The authors have answered some of the concerns, and have incorporated changes
They have however not responded to few points like about ENTREQ checklist, and incorporating ideas for future research in the discussion.

Reviewer 2 ·

Basic reporting

The authors answered the points well and revised the manuscript according to the suggestion of the reviewers.

Experimental design

No comment

Validity of the findings

No comment